# Galvanic Corrosion Due to a Heterogeneous Sulfate Reducing Bacteria Biofilm

**Hongwei Liu [1,2,][\*] , Haixian Liu [1,2] and Yuxuan Zhang [1,2]**

1 School of Chemical Engineering and Technology, Sun Yat-sen University, Zhuhai 519082, China;
liuhaixianw@163.com (H.L.); zhyx9621@163.com (Y.Z.)
2 Southern Marine Science and Engineering Guangdong Laboratory (Zhuhai), Sun Yat-sen University,
Zhuhai 519082, China
\* Correspondence: liuhw35@mail.sysu.edu.cn

**Abstract:** In this work, the galvanic corrosion behavior of sulfate reducing *Desulfotomaculum nigrificans* biofilm-covered and uncovered carbon steel was investigated using various electrochemical measurements. The results showed that the bare specimen in the abiotic solution functions as the anode; whereas the biofilm-covered specimen in the SRB-containing solution functions as the cathode after two electrodes being coupled. The anodic reaction of specimen in the biotic solution containing SRB was inhibited; whereas the cathodic reaction was considerably promoted after coupling. Hence, localized corrosion of specimen in the abiotic solution was observed due to the galvanic corrosion effect. SRB could still accelerate steel corrosion even after coupling, but the results indicate that the contribution of SRB to steel corrosion decreased. The localized corrosion of steel in the SRB-containing environments not only involved the SRB biofilm, but also a galvanic corrosion effect. The flow of electrons from the anodic dissolution of Fe in the abiotic solution to the SRB cells of cathodic area decreased the acceptance capacity of electrons by SRB from steel beneath biofilm. As a result, the steel corrosion beneath SRB biofilm decreased after coupling.

**Keywords:** microbiologically influenced corrosion; galvanic corrosion; sulfate-reducing bacteria; biofilm

## 1. Introduction

Microbiologically influenced corrosion (MIC) has been considered as one of the key reasons causing the failure of pipelines [1,2]. Sulfate reducing bacteria (SRB) have been commonly found in various environments, including oil produced water, sea water, soil, and sewage [3,4]. SRB MIC can significantly accelerate the localized corrosion against pipeline steel in the oil and gas industry [5]. The localized corrosion is the biggest threat to the safe running of pipelines. Some researchers have proposed some SRB corrosion mechanisms, including cathodic depolarization, oxygen-concentration cell, biocatalysis, etc. [6–8]. However, the localized corrosion mechanism induced by SRB is still ambiguous, because SRB corrosion environment is not only very complex but changes in natural conditions.

It has been verified that SRB MIC is closely related to the biofilm formed on the steel surface, and SRB biofilm is heterogeneous [9]. SRB biofilm is mainly composed of SRB cells, extracellular polymeric substance (EPS) and corrosion products. Although much work has been done, the mechanism of SRB MIC still cannot be fully explained. In the natural conditions, SRB biofilm forms on the steel surface heterogeneously, which means that some areas of pipeline are covered by biofilm while other areas of pipeline have only corrosion products without SRB cells. The electrochemical potential of biofilm-covered area and the abiotic corrosion product-covered area are different [10]. This means that a galvanic effect between the biofilm-covered and uncovered areas can exist. It is very important

to illustrate the galvanic corrosion behaviors and mechanism, which can be one of a key reason that leading the localized corrosion by MIC. Therefore, the similar studies can promote the further development of MIC mechanism. However, there are no reports which have investigated this possibility. The authors' previous reports [11–13] have indicated that SRB could promote the galvanic corrosion effect between the deposit-covered steel and bare steel.

The use of a wire beam electrodes (WBEs) is a useful technique, which has been widely used to study localized corrosion as well as the galvanic corrosion [14]. The distribution of current density of potential can be monitored in situ to provide more information about localized corrosion under surface film. Wu et al. [15] studied the effect of microbes on the water line corrosion processes using WEBs, and found that different anode–cathode distribution characteristics were observed in the absence and presence of microbes over time. Chen et al. [16] found that *Thalassospira sp* biofilm provided the electrochemical active sites for the initiation of pitting corrosion of Q235 carbon steel using WBEs, and the current distribution of large cathodic area and small anodic area contributed to the propagation of pitting corrosion.

In this work, the galvanic corrosion of sulfate reducing *Desulfotomaculum nigrificans* biofilm-covered and uncovered carbon steel was investigated using open circuit potential (OCP), potentiodynamic polarization curves, galvanic current densities, and measurements with WBEs. Two home-made WBEs (i.e., 6 × 10 and 4 × 10 arrays of steel electrodes) were used to monitor the distribution of current density and potential under biofilm and abiotic surface film.

## 2. Experimental

Q235 carbon steel was used in this work with a chemical composition (wt.%): C 0.30, Mn 0.42, S 0.03, P 0.01, Si 0.01, and Fe balance. The electrode with a work area of 0.785 cm$^2$ was used to do OCP and potentiodynamic polarization curve measurements. All specimens were sealed using epoxy, and were ground sequentially up to 1200 grit silicon carbide paper prior to testing. Then, the specimens were degreased and washed using acetone and anhydrous ethanol, respectively, and dried in high-purity N$_2$ (99.99%). Finally, the specimens were sanitized under an ultraviolet (UV) lamp for 30 min.

*Desulfotomaculum nigrificans* were used in this work, which were isolated from Shengli oilfield in China [17]. SRB culture medium contained (g/L) NaCl 10, yeast extract 1.0, MgSO$_4$·7H$_2$O 0.2, K$_2$HPO$_4$ 0.01, (NH)$_2$Fe(SO$_4$)$_2$ 0.2, vitamin C 0.1, and 4.0 mL/L sodium lactate (pH 7.2). SRB were incubated at 37 °C. In this work, SRB culture medium was used as the test solution, and the abiotic culture medium was used as control. SRB culture medium was autoclaved at 121 °C for 20 min, then was deaerated by purging high-purity nitrogen for 2 h.

The OCP and polarization curves were measured using an electrochemical workstation (Model CS350, Corrtest, Wuhan, China), where a saturated calomel electrode (SCE) and a platinum plate were used as a reference and counter electrodes, respectively. Potentiodynamic polarization curves were measured after OCP reached a steady state, and the potential scanning range was from −250 mV to +350 mV vs. corrosion potential ($E_{corr}$), with a sweep rate of 0.5 mV/s. The experimental setup used for electrochemical measurements were shown in Figure 1. A 0.22 μm filter film in Figure 1 was used to hinder SRB cells. Galvanic current density between the steel under biofilm and abiotic corrosion product film was measured using a zero-resistance ammeter (ZRA, Model CST500, Corrtest, Wuhan, China) in the same aeration cell. For the galvanic current density measurements, two identical electrodes were placed in the abiotic and biotic solution in Figure 1.

Two WBEs composed of 60 and 40 pieces of steel disks (1.5 mm in diameter) were used, where the 60 disks were placed in the SRB-containing test solution and the other 40 disks were placed in the abiotic test solution. The current density and potential monitoring were conducted using an electrochemical instrument (Model CST520, Corrtest, Wuhan, China). A 10 × 10 autoswitch array was installed to switch the electrical connection of individual steel disks, and each of the disks was used as the working electrode one at a time. All tests were repeated three to five times and the data points in this work and are reported with standard deviations.

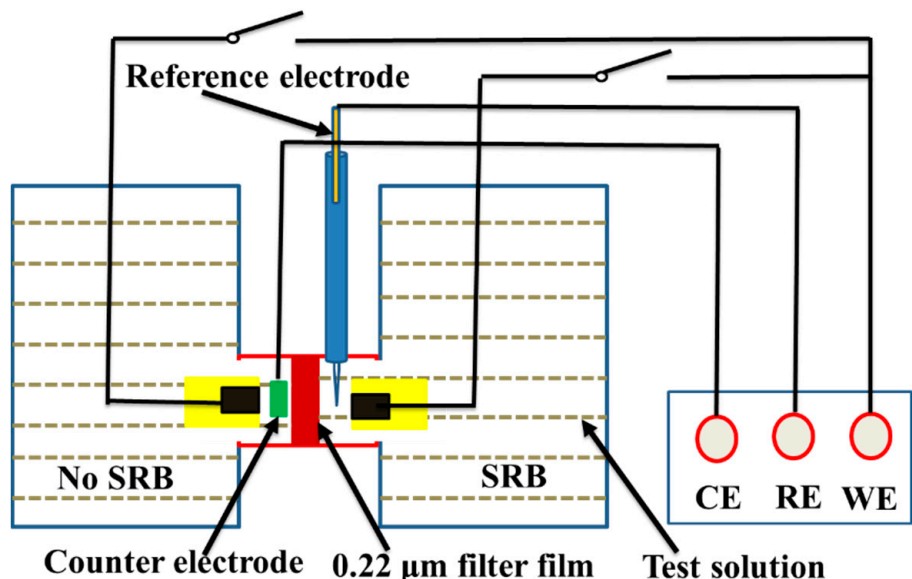

**Figure 1.** Schematic diagram of the experimental setup for electrochemical measurements.

## 3. Results

The time-dependence of the OCP of the specimens without coupling is shown in Figure 2, which shows that the OCP values of the specimen in the SRB-containing solution were more positive than those in the abiotic solution. This means that the specimen in the abiotic solution is as the anode; whereas the specimen in the SRB-containing solution is as the cathode. Both the OCPs in the abiotic and SRB-containing solution had an increase during the 4 testing days, then decreased gradually with time until the tenth day. OCPs reached a steady state after 10 days of testing. The positive OCP in the SRB-containing solution can be due to the acceleration of the cathodic reaction by SRB [18].

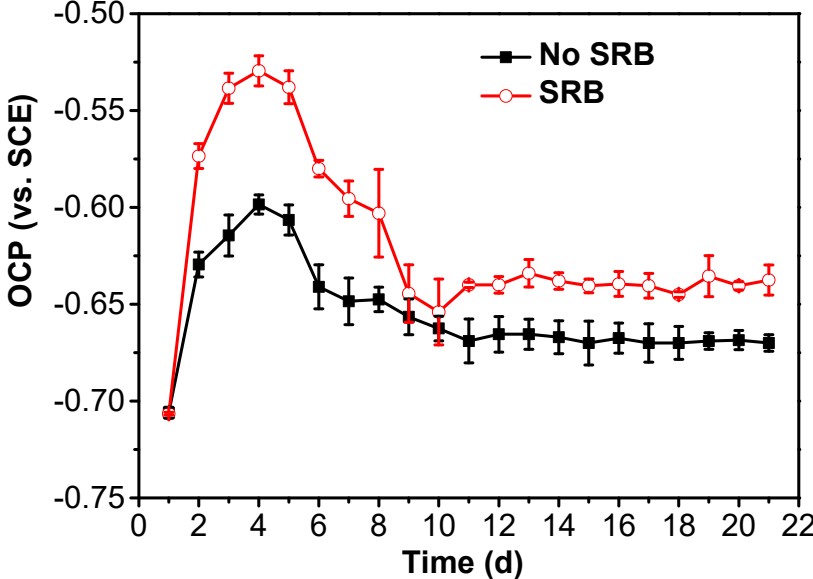

**Figure 2.** Time dependence of OCP of specimens in the test solution in the absence and presence of SRB.

Figure 3 shows the potentiodynamic polarization curves of specimens in the abiotic and SRB-containing test solution without and with coupling after 21 days of testing. Under the coupled condition, after 21 days, these two electrodes were disconnected and then the polarization curves were measured under coupled and uncoupled conditions. It is seen that both the anodic and cathodic

reactions were considerably accelerated by SRB, which caused a higher corrosion current density (Figure 3a). Under the coupled condition, it is seen that the anodic reaction of specimen in the abiotic test solution was accelerated while the cathodic reaction was inhibited compared with that with no coupling (Figure 3a,b). The passivation of anodic branch in the SRB-containing solution could be observed (Figure 3b). The anodic reaction of specimen in the SRB-containing solution was inhibited while the cathodic reaction was considerably promoted after coupling compared to the case with no coupling (Figure 3a,b).

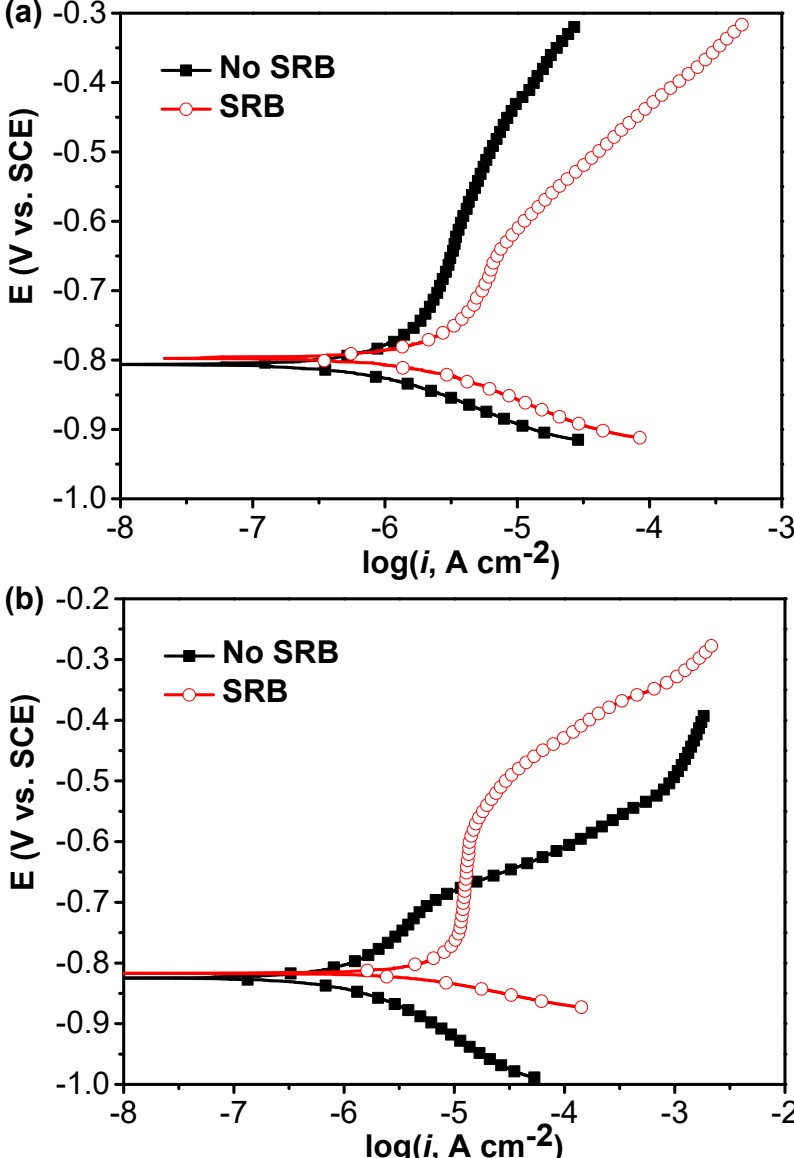

**Figure 3.** Potentiodynamic polarization curves of specimens in the abiotic and SRB-containing test solution without (**a**) and with (**b**) coupling after 21 days of testing.

The change of galvanic current densities of the specimens in the abiotic and SRB-containing solution with time are shown in Figure 4. The galvanic current densities are observed to decrease during the initial 2 days, then have an apparent increase after 2 days of testing. The galvanic current densities kept a steady state after 2 days of testing until the tenth day. After 10 days of testing, the galvanic current densities had an abrupt decline, and the values of the galvanic current densities

were negligible. This suggested that the galvanic effect was very small and could be neglected after 10 days of testing.

Figures 5 and 6 shows the changes in distribution of current density and potential of WBEs with time in the abiotic and SRB-containing solution. There was a peak of anodic current density that could be observed at electrode 84 in the SRB-containing solution during the initial 7 days of testing (Figure 5). A weak peak of anodic current density also could be seen at electrode 7 in the abiotic solution in the initial 7 days of testing (Figure 5). There was a large potential difference in the initial 7 days (Figure 6). These results suggested that there was a high galvanic effect between specimens in the abiotic and SRB containing solution. Furthermore, the galvanic effect accelerated the localized corrosion of specimen in the abiotic solution. Even so, after coupling, some local anodic sites still could be found in the SRB-containing solution, which further verified that SRB could directly promote steel corrosion.

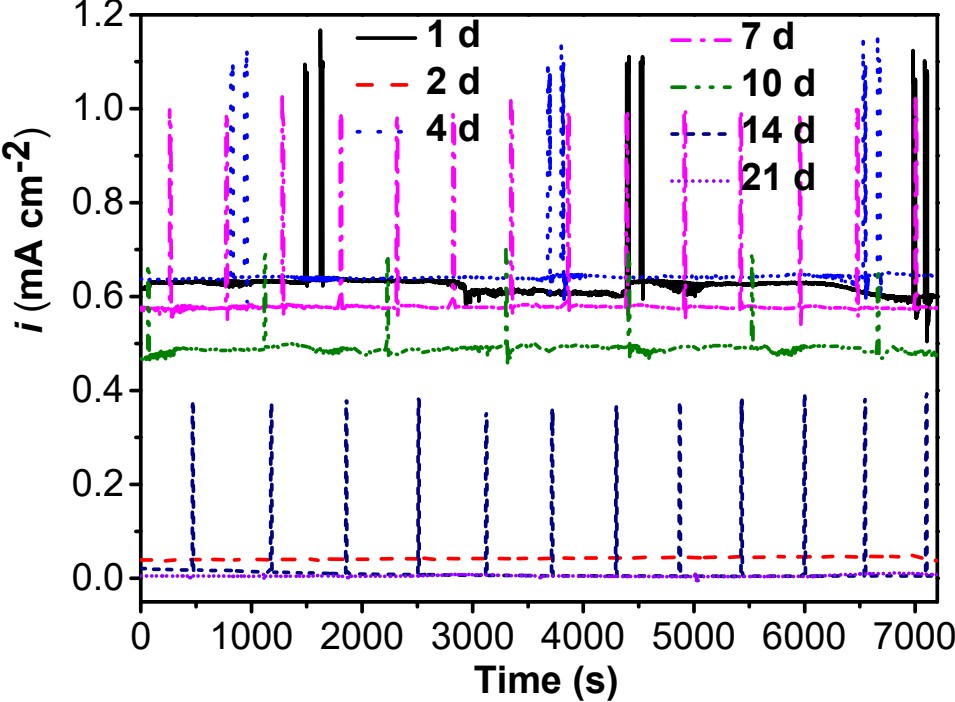

**Figure 4.** The change of galvanic current densities of specimens in the abiotic and SRB-containing test solution with time.

After 7 days of testing, both the anodic sites changed, a new peak of anodic current density at electrode 10 could be found (Figure 5). Some new peaks of anodic current density with smaller values in the abiotic and SRB-containing solution could be seen after 10 days of testing, indicating that the galvanic effect is weak (Figure 5). After 10 days of testing, the differences of potential were much smaller with values of about 1 mV (Figure 6), which also indicated there was a weak galvanic effect for the specimen in the abiotic and SRB-containing test solution. The distributions of current density in Figure 5 correspond to the result of potential distribution in Figure 6.

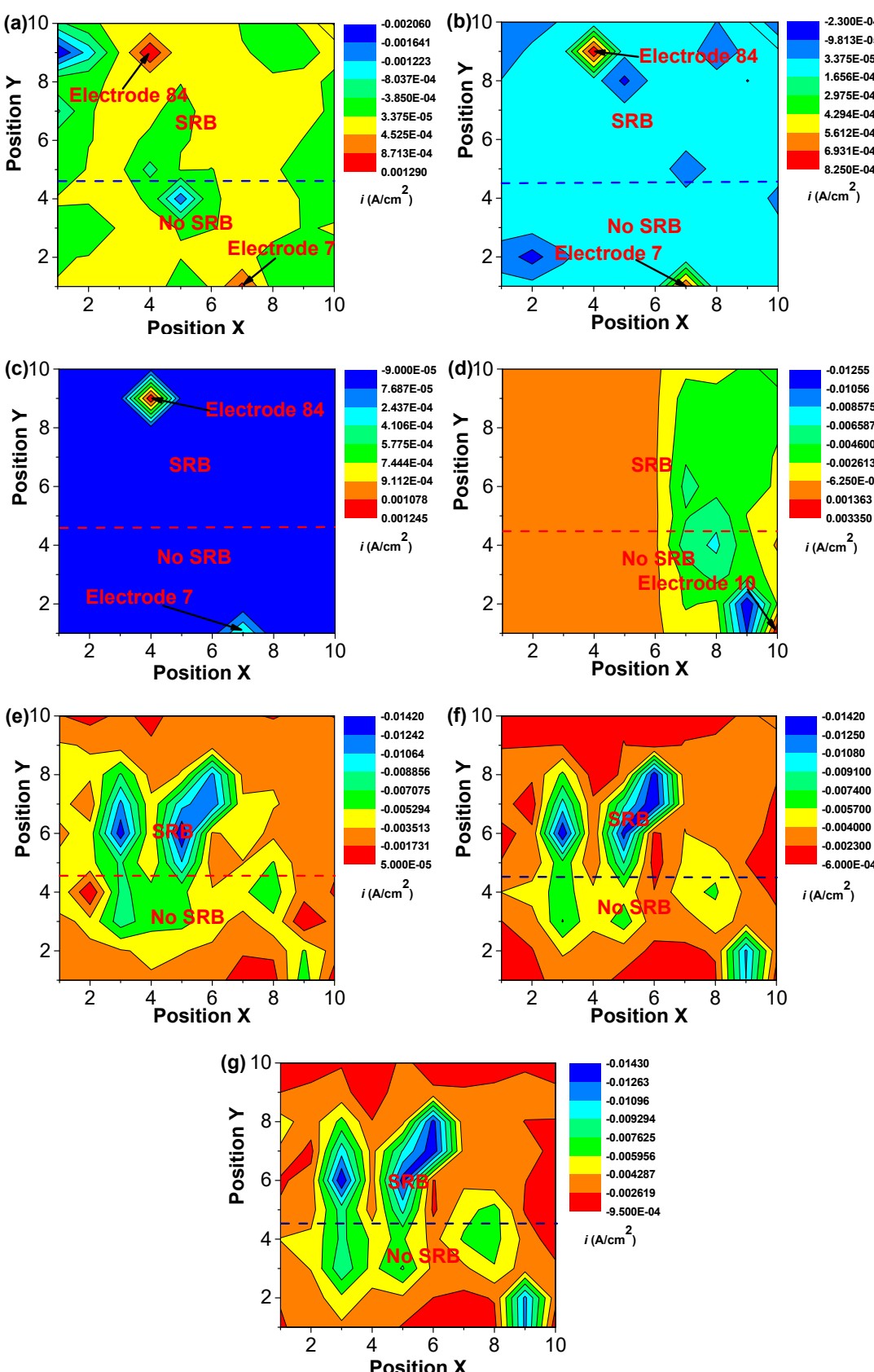

**Figure 5.** The changes of distribution of current density of WBEs with time in the abiotic and SRB-containing test solution: (**a**) 1 d; (**b**) 2 d; (**c**) 4 d; (**d**) 7d; (**e**) 10 d; (**f**) 14 d; (**g**) 21 d.

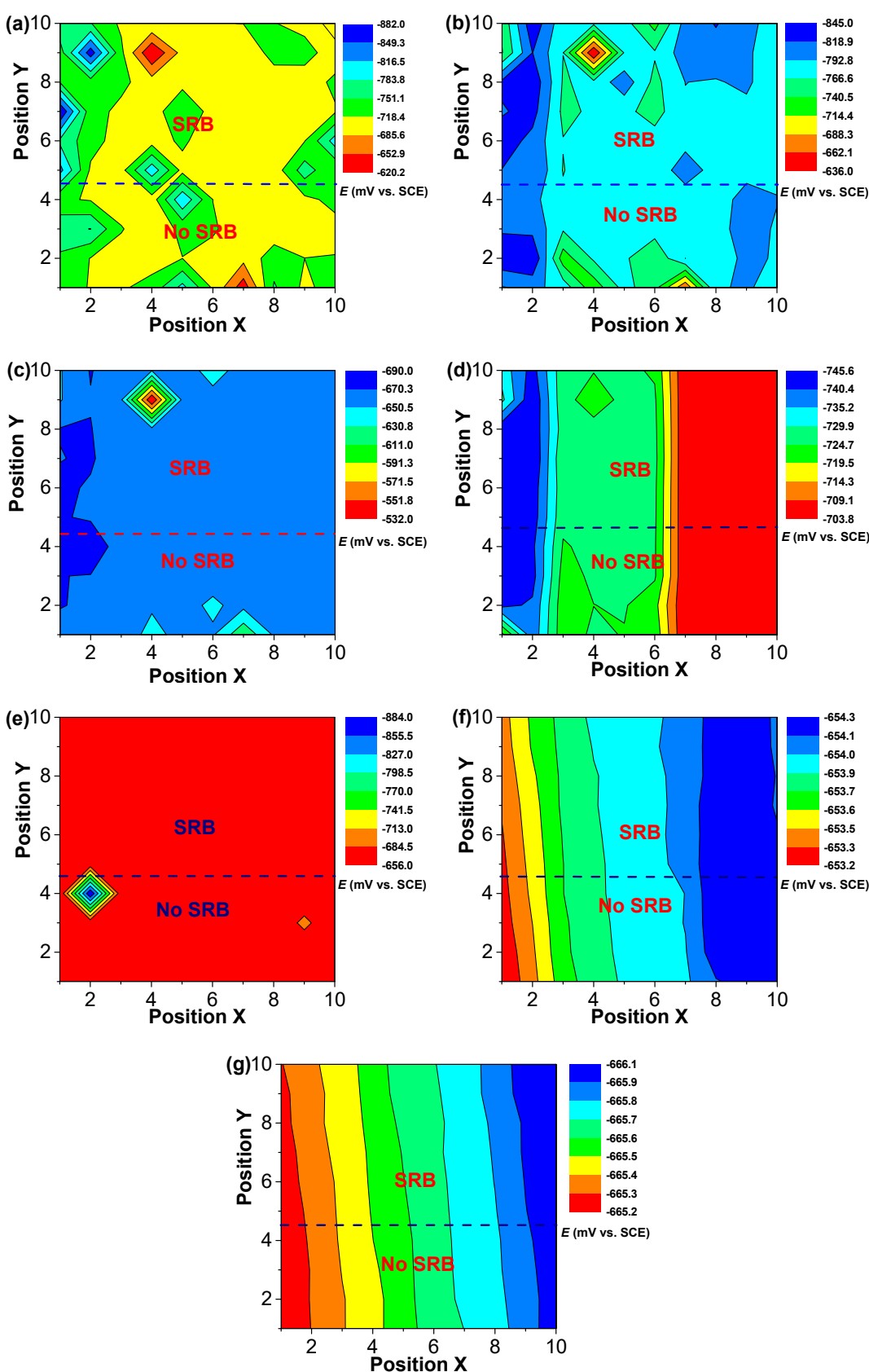

**Figure 6.** The changes of distribution of potential of WBEs with time in the abiotic and SRB-containing test solution: (**a**) 1 d; (**b**) 2 d; (**c**) 4 d; (**d**) 7d; (**e**) 10 d; (**f**) 14 d; (**g**) 21 d.

## 4. Discussion

The electrochemical corrosion behaviors were done using a dual-cell. The above results (Figures 2–6) have shown that there was a galvanic effect between the specimens in the abiotic and SRB-containing solution, where the specimen in the abiotic solution acts as the anode and the specimen in the SRB-containing solution acts as the cathode. Lopes et al. [19] also used a dual-electrochemical cell to study the biofilm of stainless steel, and found that a transfer of electrons from the stainless-steel sample of the anaerobic cell to the stainless-steel sample of the aerobic one. The previous reports have indicated that SRB can grow well in the test solution, and the total growth period was about 14 days [17,20]. The galvanic effect can promote the localized corrosion of the specimen in the abiotic test solution (Figure 5). Most previous reports [21,22] have verified that SRB biofilm could considerably accelerate steel corrosion, especially for localized corrosion, and this work also found that SRB could accelerate steel corrosion without coupling (Figure 3a). This work firstly verified that the galvanic effect could accelerate localized corrosion of steel in the abiotic solution, which adds further to the possible localized corrosion mechanisms in the SRB-containing environments. Figure 3b shows that cathodic reaction, i.e., the reduction in sulfate, was considerably accelerated after coupling in the SRB-containing solution, and the anodic reaction was inhibited. However, the corrosion current density of the specimen in the SRB-containing solution was still higher than that in the abiotic test solution. Taken together, these results demonstrate that SRB MIC could still accelerate steel corrosion even after coupling. However, the contribution of SRB MIC to steel corrosion decreased after coupling (Figure 3b). The WBE results also show that the localized anodic cites could still be found in the presence of SRB, which further verified that SRB biofilm could indeed accelerate steel corrosion after coupling.

The galvanic current density changed with time and the galvanic current density was small after 10 days of testing (Figure 5). Zheng et al. [20] found that SRB cell counts had an abrupt decrease after 10 days of testing in the SRB culture medium. After 10 days of testing, the galvanic effect could be neglected due to a smaller galvanic current density current (Figure 5). These results suggest that the galvanic effects could be related to SRB activity. The galvanic effect was closely related to the potential difference, and a larger potential difference would cause a higher galvanic effect [11]. Figures 2 and 6 have shown that the potential differences turned out to be much smaller after 10 days of testing, which directly leads to an abrupt decrease in galvanic effect. Some previous reports [17,23] have shown that SRB biofilm only promoted the positive shift of potential at initial test time. Similar results also were found in this work (Figure 2). In this work, the two cells were separated by a 0.22 μm filter film. A FeS film would form on the specimen surface in the abiotic solution due to the diffusion of sulfides. FeS film could have a protective effect that promotes the positive shift of potential (Figure 2). SRB MIC was a dynamic process. A similar corrosion product film could be formed on the two-specimen surface after a long time, leading to the decrease in potential difference. In addition, the galvanic effect also decreased after 10 days of testing.

A mechanistic model illustrating galvanic corrosion is shown in Figure 7. Some anodic sites could form on the specimen surface in the abiotic and SRB-containing solution under the coupled condition. Even so, localized corrosion still could form beneath SRB biofilm under the coupled condition. The electrons from the anodic dissolution could flow to the cathodic area used to the reduction in sulfate. SRB cells in the biofilm could also directly obtain electrons from Fe [24,25] and use them for the reduction in sulfate. As a result, the cathodic reaction of SRB MIC was considerably accelerated. Some SRB cells in the biofilm can also get electrons from the anodic dissolution in the abiotic test solution, so the capacity to accept electrons of SRB from the dissolution of Fe beneath biofilm would decrease. This would directly cause the decrease in SRB MIC contribution to steel corrosion (Figures 2 and 5). However, the localized corrosion induced by SRB were still be quite serious (Figure 5).

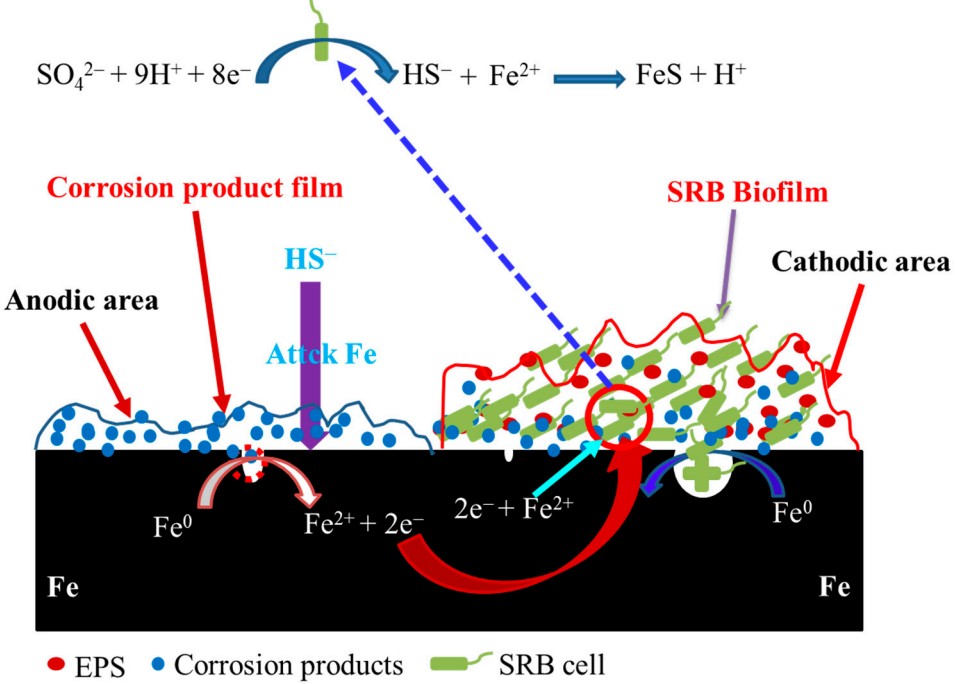

$$SO_4^{2-} + 9H^+ + 8e^- \qquad HS^- + Fe^{2+} \longrightarrow FeS + H^+$$

**Figure 7.** A mechanistic model illustrating galvanic corrosion behavior of sulfate reducing *Desulfotomaculum nigrificans* biofilm-covered and uncovered carbon steel, where EPS refers to extracellular polymeric substances. Initially the specimen in the abiotic solution is as the anode, and some local anodic sites can be found on the specimens' surface in the abiotic and SRB-containing test solution.

## 5. Conclusions

There was a galvanic effect of specimens in the abiotic solution and SRB-containing solution, in which the specimen in the abiotic solution acts as the anode and the specimen in the SRB-containing solution acts as the cathode. The anodic reaction of the specimen in the SRB containing solution was inhibited; whereas the cathodic reaction was considerably promoted after the two electrodes were coupled. The passivation of anodic branch in the SRB-containing solution could be observed after coupling. The galvanic effect accelerated the localized corrosion of specimen in the abiotic test solution. The localized corrosion beneath biofilm was still more serious than that in the abiotic test solution after coupling. The flow of electrons from the anodic dissolution of Fe in the abiotic solution to the SRB cells of cathodic area decreased the acceptance capacity of electrons by SRB from the dissolution of Fe beneath biofilm. As a result, the steel corrosion beneath SRB biofilm decreased after coupling.

**Author Contributions:** Conceptualization, H.L. (Hongwei Liu); Formal analysis, H.L. (Haixian Liu); data curation, H.L. (Hongwei Liu) and Y.Z.; writing—review and editing, H.L. (Hongwei Liu), H.L. (Haixian Liu), and Y.Z. All authors have read and agreed to the published version of the manuscript.

**Funding:** This research was funded by the Fundamental Research Funds for the Central Universities (No. 19lgzd18), National Natural Science Foundation of China (No. 51901253), Guangdong Basic and Applied Basic Research Foundation (No. 2019A1515011135) and the Open Project Program of Beijing Key Laboratory of Pipeline Critical Technology and Equipment for Deepwater Oil & Gas Development (No. BIPT201904).

**Conflicts of Interest:** The authors declare that they have no conflicts of interest.

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
