# Peer review of "Galvanic Corrosion Due to a Heterogeneous Sulfate Reducing Bacteria Biofilm"

_coatings, doi:10.3390/coatings10111116_

Round 1
Reviewer 1 Report
Dear authors,
thank you for the interesting paper.
My comments are only:
- line 52 - Chen et al. [16] - the reference [16] is Kodama et al.,
- line 183 - A mechanistic model illustrating galvanic corrosion is shown ... (not was shown).
Best regards.

Author Response
line 52 - Chen et al. [16] - the reference [16] is Kodama et al.,
Thanks for your comments. Correction has been done as suggested.
line 183 - A mechanistic model illustrating galvanic corrosion is shown ... (not was shown).
Thanks for your comments. Correction has been done as suggested.
Reviewer 2 Report
It is a very good paper, clearly written. Only one information is missing to support the data interpretation and the conclusions: nothing is said about the growth of bacteria in the test solution where SRBs are added. Did they grow? The authors need to include some information about the development of bacteria during their experiments, It could be the evolution of bacteria numbers, of sulphide concentrations (or the decrease of sulphate concentration), the evolution of optical density of the test solution, ....
It the reason why I ask for a revision of the paper.
Two additionnal comments:
1- Regarding the figure 7: the red arrow of electrons going from the metal to the biofilm seems to indicate that the electrons flow througth the corrosion products, the solution and the biofilm. Is it correct? if yes, some explainations wil be welcome, if no, thank you to modify this red arrow.
2-The second comment is regarding the references: quite a lot of experiments with dual cells as presented in figure 1 as been done in the past. I suggest to add one or two references from these works and not only one self-citation.
Author Response
It is a very good paper, clearly written. Only one information is missing to support the data interpretation and the conclusions: nothing is said about the growth of bacteria in the test solution where SRBs are added. Did they grow? The authors need to include some information about the development of bacteria during their experiments, It could be the evolution of bacteria numbers, of sulphide concentrations (or the decrease of sulphate concentration), the evolution of optical density of the test solution, ....
It the reason why I ask for a revision of the paper.
Thanks for your comments. Actually, the growth curves of SRB have been done in our previous reports. And we have cited. Liu H , Fu C , Gu T , et al. Corrosion behavior of carbon steel in the presence of sulfate reducing bacteria and iron oxidizing bacteria cultured in oilfield produced water. Corrosion Science, 2015, 100:484-495. Zheng B , Li K , Liu H , et al. Effects of Magnetic Fields on Microbiologically Influenced Corrosion of 304 Stainless Steel[J]. Industrial & Engineering Chemistry Research, 2013, 53(1):48–54.
Two additionnal comments:
1- Regarding the figure 7: the red arrow of electrons going from the metal to the biofilm seems to indicate that the electrons flow througth the corrosion products, the solution and the biofilm. Is it correct? if yes, some explainations wil be welcome, if no, thank you to modify this red arrow.
Thanks for your comments. The electrons flow through the Fe not corrosion products, the solution and the biofilm. And the red arrow has been revised.
2-The second comment is regarding the references: quite a lot of experiments with dual cells as presented in figure 1 as been done in the past. I suggest to add one or two references from these works and not only one self-citation.
Thanks for your comments. References 19 has been added according to the suggestion.
Reviewer 3 Report
Overall opinion
- structure good
- length appropriate
- goal clearly formulated
- used analysis/measurement methods relevant and adequate
- scientific importance average
- English good, with some few style issues
- results credible and consistent
Suggestions:
- The authors might consider elaborating a few lines about the expected consequences, importance, or even possible applications of their results. In other words: why is it useful to learn about the galvanic corrosion process between the SRB-biofilm-covered and non-biofilm-covered steel? What can we gain from these results?
- The legends of Fig. 6 are too small; enlarging them would help the reader.
Author Response
The authors might consider elaborating a few lines about the expected consequences, importance, or even possible applications of their results. In other words: why is it useful to learn about the galvanic corrosion process between the SRB-biofilm-covered and non-biofilm-covered steel? What can we gain from these results?
Thanks for your comments. Correction has been done as suggested in the page 2.
The legends of Fig. 6 are too small; enlarging them would help the reader.
Thanks for your comments. Correction has been done as suggested.